**Data Availability Statement:** Python Code is available at www.github.com/gerritgr/Covid19Dispersion.

**Funding:** This work was partially supported by the DFG project MULTIMODE (391984329). There was

# Heterogeneity matters: Contact structure and individual variation shape epidemic dynamics

**Gerrit Großmann**⬮*, **Michael Backenköhler, Verena Wolf**

Saarland Informatics Campus, Saarland University, Saarbrücken, Germany

* gerrit.grossmann@uni-saarland.de

## Abstract

In the recent COVID-19 pandemic, mathematical modeling constitutes an important tool to evaluate the prospective effectiveness of non-pharmaceutical interventions (NPIs) and to guide policy-making. Most research is, however, centered around characterizing the epidemic based on point estimates like the average infectiousness or the average number of contacts. In this work, we use stochastic simulations to investigate the consequences of a population's heterogeneity regarding connectivity and individual viral load levels. Therefore, we translate a COVID-19 ODE model to a stochastic multi-agent system. We use contact networks to model complex interaction structures and a probabilistic infection rate to model individual viral load variation. We observe a large dependency of the dispersion and dynamical evolution on the population's heterogeneity that is not adequately captured by point estimates, for instance, used in ODE models. In particular, models that assume the same clinical and transmission parameters may lead to different conclusions, depending on different types of heterogeneity in the population. For instance, the existence of hubs in the contact network leads to an initial increase of dispersion and the effective reproduction number, but to a lower herd immunity threshold (HIT) compared to homogeneous populations or a population where the heterogeneity stems solely from individual infectivity variations.

## Introduction

At the beginning of 2020, the world was struck by the coronavirus (SARS-CoV-2) pandemic. Faced with the approaching overload of healthcare systems, the international community turned to non-pharmaceutical interventions (NPIs) in an attempt to contain the spread of the pathogen [1]. Computational epidemiological modeling became an important asset to predict the propagation and to evaluate the prospective effectiveness of various measures such as school closings and travel restrictions [2, 3]. For an overview of COVID-19 models and their successes and failures, we refer the reader to [4, 5]. Literature abounds with new studies describing and forecasting the spread of COVID-19. Instead, we focus on fundamental properties of popular models and the consequences of popular modelling assumptions.

In this work, we highlight the importance of population heterogeneity for computational epidemiology and explain why many models used in the current COVID-19 pandemic do not adequately capture it. In particular, we analyze how individual variations in contact numbers

no additional external funding received for this study.

**Competing interests:** The authors have declared that no competing interests exist.

and infectiousness influence the evolution of a pandemic. We use the term *infectiousness* as a property of the host to denote the probability of passing a pathogen, given an established contact to a susceptible individual. We qualitatively study the dynamical evolution based on different properties like the height of the infection peak and the fluctuations of the *effective reproduction number $R_t$* (average number of secondary infections at time point *t*). Furthermore, we study how this heterogeneity influences the *dispersion* during an epidemic's evolution. COVID-19 is associated with an exceptional high dispersion and understanding how it emerges is a crucial asset in controlling the pandemic [6].

A noticeable example of heterogeneity in a population's interaction structure are individuals with extraordinary many contacts, so-called *hubs*. Similarly, super-spreader events refer to temporary gatherings where one infected individual (potentially) infects many others. Early on in the pandemic, it was pointed out that hubs are not accurately captured by many models [7]. The evidence for the importance of hubs and super-spreader events became increasingly conclusive over time [8–11].

The concept of *(over-)dispersion* is closely related [12–14] and is consistently reported for the COVID-19 pandemic [15–19]. In short, this concept reflects that a small number of infected individuals infect many others while most infected individuals infect no one or only very few. Overdispersion can be caused by hubs (many contacts, average transmission probability) but also by individuals with high infectiousness (average contact number, high transmission probability). Different individual levels of infectiousness are a source of further heterogeneity in the population.

Viral load levels (and other properties that determine a host's infectiousness) differ between individuals and within individuals over time [20–23]. While many models include the temporal aspect, the effects of individual variations are not well explored. Moreover, some virus variants are associated with higher infectiousness [24].

In order to study the effects of heterogeneity, we translate an ODE model for the spread of COVID-19 to a stochastic network-based model. More specifically, this work uses contact networks (i.e., graphs) to model individual variations in the number of social contacts. To model the individual viral load variations, we use a randomly drawn infectiousness parameter for each individual. Epidemic spreading on networks is well understood and contact networks are a universal and well-established paradigm for the analysis of complex interaction structures.

This paradigm allows us to study population heterogeneity while keeping population averages fixed. In particular, we only compare networks with the same connectivity (i.e., number of edges). We also fix the mean infectiousness (i.e., we only modulate how much an individual may deviate). On this premise, we investigate the effects of population heterogeneity on the emergence of dispersion and the dynamical evolution of the epidemic. We find, for instance, that power-law networks admit a natural early growth of $R_t$ and a very high dispersion. Individual differences in infectiousness increase the dispersion even more while they generally weaken the epidemic, e.g., in terms of infection peak height and final epidemic size.

Our contributions are as follows:

1. We give an overview of popular COVID-19 models based on ODEs, branching processes and networks and discuss their (implicit) assumptions about a population's heterogeneity.

2. We show that imposing common interaction structures (i.e., using a graph to determine how infections can propagate) drastically changes an epidemic's evolution.

3. We analyze the additional effects of individual viral load variations.

4. We propose a novel method to quantify time-dependent dispersion based on an empirical analysis of simulation runs.

Contribution (2) is based on preliminary results that were published during the first COVID-19 wave in the spring of 2020 [25].

### Organization

Our manuscript is structured as follows: We present relevant literature in Section *Related work*. In Section *Method*, we show how to translate ODE models to network-based models and discuss their relationship. Section *Experiments* provides numeral experiments based on synthetically generated random contact networks. *Conclusions* completes the manuscript.

### Related work

Mathematical modeling of epidemics is a wide research area to control, predict, and understand epidemics or similar types of propagation processes (like opinions, or computer viruses). Here, we mostly focus on the network-based spreading paradigm [26] and its relation to other model types. In particular, we study which types of population heterogeneity can be expressed and how models are used in the current COVID-19 pandemic. Note that currently all models that study COVID-19 quantitatively suffer from the poor quality of data and uncertainty about parameters [27, 28]. We focus on the three model types we consider most relevant for epidemiological modeling in general. While ODE models and network-based models are directly part of the manuscript, we include branching processes because they are the de facto standard for formalizing and studying dispersion in epidemics. For a comparative analysis of models specific to COVID-19, we refer the reader to [4, 29].

### ODE models

The most wide-spread epidemiological model type is based on a system of ordinary differential equations (ODEs) in which coupled fractions of individuals in disease compartments change deterministically and continuously over time [30]. For an overview on various applications, we refer the reader to [31–34]. Compartments refer to different disease stages (e.g., susceptible (S), infected (I), recovered (R), exposed (E), dead (D)). Most commonly used is the three-compartment SIR-model (cf. Fig 1). Note that ODE models use a single parameter ($\lambda$) to model the chance to meet someone (interaction structure) and the probability to transmit the infection.

**Population heterogeneity.** Expressing population heterogeneity is only possible to a very limited degree. The typical way is to introduce additional compartments that encode a membership to a certain group (e.g., *susceptible* and "*younger than 20*"). These extended models are

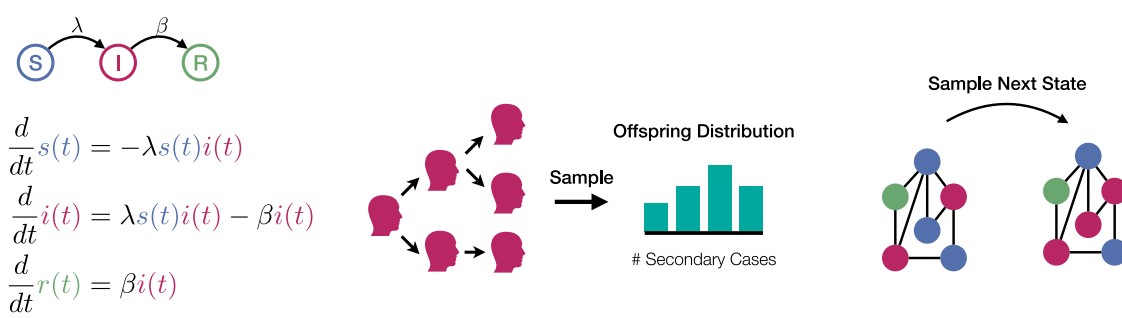

**Fig 1. Schematic overview of model types.**

often referred to as *meta-population* [35] models. Apart from that, a homogeneous interaction structure is assumed. Effects like super-spreaders (or overdispersion) are practically not expressible. The same holds for local die-outs. Moreover, the deterministic nature makes it difficult to conceptualize risk and uncertainty. ODE models arise as the mean-field limit of a well-mixed Markov population model (corresponding to a complete graph in the network-based paradigm) [36].

**COVID-19.**  Literature abounds with ODE-based COVID-19 models, some methods and applications are summarized in [37, 38]. For instance, Dehning et al. [39] use a model, where the infection rate may change over time to predict a suitable time point to loosen NPIs in Germany. José Lourenço et al. infer epidemiological parameters [40]. Khailaie et al. analyze how changes in the reproduction number affect the epidemic dynamics [41]. Stutt et al. evaluate the effectiveness of NPIs [42]. The spread of COVID-19 was studied for many more countries and settings [43–47]. Other studies use a meta-population approach and group individuals based age [48–51] or region [52–54]. Moreover, [55, 56], modify ODE models to account for individual variation in susceptibility.

Roda et al. use an ODE model to illustrate the general difficulty of predicting the spread of COVID-19 data [57]. Limitations in the applicability of ODE models regarding data from Italy is reported in [58]. Similar results are found by Castro et al. using COVID-19 data from Spain [59]. General concerns are articulated in [60].

## Branching processes

Stochastic branching processes operate in discrete or continuous-time and are useful when studying the underlying stochastic nature of an epidemic. They are based on a tree that grows over time and represents the infected individuals. The children (offspring) of each node represent an individual's secondary infections and the number of children is drawn from an *offspring distribution* with mean $R_0$ that is provided by the modeler [14, 61–63].

**Population heterogeneity.**  The offspring distribution makes it straight-forward to encode individual variations in infectiousness or connectivity. The paradigm allows to study random extinction probabilities of the epidemic and the effects of super-spreaders/overdispersion [12]. However, branching processes do not admit a (model intrinsic) saturation due to growing immunity in the population. Moreover, the high level of abstraction makes it difficult to study the effects of NPIs and the characteristics of the spatial diffusion of the pathogen.

**COVID-19.**  Zhang et al. use a branching process to measure the dispersion of COVID-19 inside China [64] and Endo et al. estimate the dispersion based on local clusters outside China [16]. Moreover, [65], use a branching process to infer epidemiological values and [22] study the influence of temporal viral load variation. Alternative branching process models to study dynamical properties specific to COVID-19 were proposed in [66–68].

## Network-based models

Network-based epidemic models use graphs to express the interactions (edges) among individuals (nodes). They are stochastic in nature and can be formulated in discrete or continuous time. Each node occupies a compartment (node state) at each point in time and infected nodes can (randomly) infect their susceptible neighbors [26, 69, 70]. Generalizations to multi-layer and weighted networks have been suggested [71].

**Population heterogeneity.**  The network-based paradigm decouples the connectivity of the population from the infectiousness of the virus. Moreover, each individual is represented by an autonomous agent which adds flexibility and makes it straightforward to include individual variations of the population. The key advantage of networks is that they represent a

universal way of encoding different types of complex interaction structures like hubs, commu-
nities, households, small-worldness, different mixing within in population-groups, etc. The
contact network can also represent spatial or geographical constraints. Network-based models
relate to ODE models in the sense that the ODE model represents the mean propagation of an
epidemic on an infinite complete graph (all nodes are directly connected), assuming that all
nodes are attributed with exponentially distributed jump times. Conceptually, the complete-
ness "removes" the heterogeneity from the interaction structure and the infinite size prevents
artifacts from the stochasticity.

**COVID-19.** Effects of different contact networks were studied in [25, 72–74]. Contact
networks are being used to build realistic simulations of a society, for instance by creating
household-structures with various types of inter-household connections [75–78]. The flexibil-
ity to control networks modeling NPIs easy [76, 77, 79, 80]. Moreover, [81–83], use a network-
based approach for spatial properties (e.g., flow between geographical regions). Although the
importance of hubs was recognized very early [84], the concrete relation to overdispersion as it
is studied in branching processes remains under-explored. Networks, where the contact struc-
ture changes over time, are particularly well-suited to study quarantine measure and social dis-
tancing [85–87].

## Method

In this section, we show how to translate a ODE model to a network-based model in order to
impose variation in connectivity and infectiousness while keeping the population averages of
clinical and transmission parameters fixed. We use a COVID-19 ODE model that is heavily
inspired by the SIR-extension of [76]. A summary of the model is depicted in Fig 2 and
Table 1. We do note upfront that we are only interested in qualitative results and do not rely
on exact parameter values.

## ODE model

Our model contains six disease stages or compartments (cf. Fig 2): *susceptible* (S), *exposed* (E)
(infected but not yet infectious), *removed* (R) (immune or dead), as well as *mild*, *severe*, and
*critical* infection stages ($I_1$, $I_2$, $I_3$). In contrast to [76], we merge dead and recovered stages to
a single *removed* stage, as both do not influence the infection dynamics further (we assume
immunity after recovery. Note that perfect and permanent immunity is not given for COVID-
19. In this study, we ignore the impact of re-infected individuals. The fraction of individuals in
each compartment evolves according to a system of ordinary differential equations (ODEs)
given in Fig 2b. Unlike network-based models, ODE models admit a semantics that is invari-
ant to the population size. Thus, we assume that the population is normalized. A further differ-
ence to [76] is that we only have a single infection parameter $\gamma$. All other parameters have a
meaningful clinical interpretation and can be specified accordingly (cf. Table 1). The set of
transition parameters $\gamma$, $\mu_j$, $\beta_j$ gives raise to a specific $R_0$. We can compute $R_0$ by assuming
that an infinitesimal fraction $\epsilon$ (representing patient zero in an infinitely large population) is
infected ($I_1$) and that the rest of the population ($1 - \epsilon$) is susceptible. Specifically, $R_0$ is the
ratio between $\epsilon$ and the population fraction that leaves S due to $\epsilon$. Therefore, we consider the
outflow from S to E caused by this initially infected fraction while it passes the three disease
stages (taking into consideration that only an even smaller fraction of $\epsilon$ reaches $I_2$ and $I_3$):

$$R_0 = \frac{\gamma}{\mu_2 + \beta_1} + \frac{\mu_2}{\mu_2 + \beta_1} \cdot \frac{\gamma/z}{\beta_2 + \mu_3} + \frac{\mu_2}{\mu_2 + \beta_1} \cdot \frac{\mu_3}{\mu_3 + \beta_2} \cdot \frac{\gamma/z}{\beta_3} \ .$$

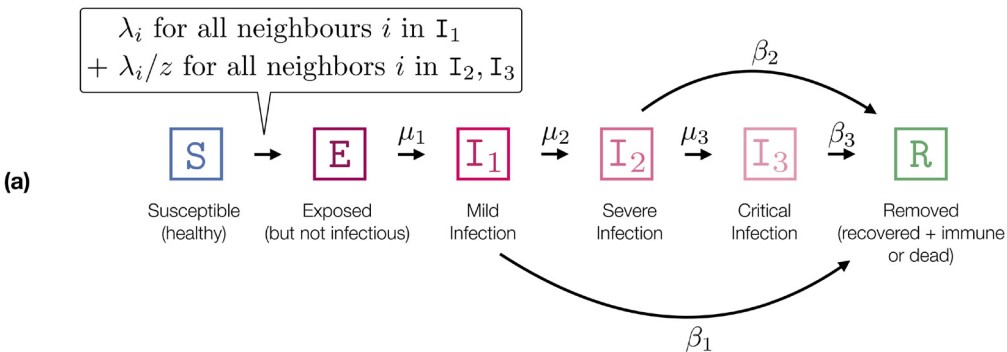

$$\frac{d}{dt}s(t) = -s(t)\left(\gamma i_1(t) + \frac{\gamma}{z}i_2(t) + \frac{\gamma}{z}i_3(t)\right) \qquad \frac{d}{dt}i_2(t) = \mu_2 i_1(t) - \mu_3 i_2(t) - \beta_2 i_2(t)$$

$$\frac{d}{dt}e(t) = s(t)\left(\gamma i_1(t) + \frac{\gamma}{z}i_2(t) + \frac{\gamma}{z}i_3(t)\right) - \mu_1 e(t) \qquad \frac{d}{dt}i_3(t) = \mu_3 i_2(t) - \beta_3 i_3(t)$$

$$\frac{d}{dt}i_1(t) = \mu_1 e(t) - \beta_1 i_1(t) \qquad \frac{d}{dt}r(t) = \beta_1 i_1(t) + \beta_2 i_2(t) + \beta_3 i_3(t)$$

**Fig 2. COVID-19 model. (a)** Transitioning system of the network model with subject-level infectiousness ($\lambda_i$ for subject $i$). The transition rates refer to exponentially distributed residence times. The expected residence time in each disease stage is the inverse of the sum of the outgoing transitions (e.g. for $\mathtt{I}_1$ it is $1/(\beta_1 + \mu_2) = 6$ (days)). Likewise, the probability to go to $\mathtt{I}_2$ is 0.2.). **(b)** Corresponding ODE model with infection rate $\gamma$, where $\gamma$ encodes connectivity and infectiousness.

For instance, $\frac{\mu_2}{\mu_2+\beta_1}$ refers to the fraction of $\epsilon$ that reaches $\mathtt{I}_2$ and $\frac{\gamma/z}{\beta_2+\mu_3}$ corresponds to the out-flow of $\mathtt{S}$ attributed to this fraction.

Hence, we can fix $R_0$ and thereby control $\gamma$. We use $R_0 = 2.5$ which leads to $\gamma \approx 0.394$.

## Network-based dynamics

We consider a static, undirected, unweighted, strongly connected graph. At each point in time, each node (individual) is annotated with a compartment. The dynamics is specified as a

**Table 1. Model parameters.**

| Parameter | Value | Meaning |
|---|---|---|
| $\lambda$ | 0.0706 | Infection rate when fixed (for $R_0 = 2.5$ and $k_{\mathrm{mean}} = 8$). |
| $\lambda_i$ | – | Infection rate (when variable) of subject $i$, $\lambda_i \sim \nu$ |
| $\nu$ | – | Density of $\lambda_i$ with $E[\lambda_i] = \lambda$. E.g., $\nu = \mathrm{Exp}(\lambda)$. |
| $z$ | 5.0 | Reduction in infectivity in disease stages $\mathtt{I}_2$, $\mathtt{I}_3$ |
| $R_0$ | 2.5 | Basic reproduction number (for fixed $\lambda$). |
| $k_{\mathrm{mean}}$ | $\approx 8$ | Mean number of neighbors (by construction). |
| $\mu_1$ | 1/5 | Disease progression rate in $\mathtt{E}$ |
| $\mu_2$ | 0.2/6 | Disease progression rate in $\mathtt{I}_1$ |
| $\mu_3$ | 0.25/6 | Disease progression rate in $\mathtt{I}_2$ |
| $\beta_1$ | 0.8/6 | Recovery rate in $\mathtt{I}_1$ |
| $\beta_2$ | 0.75/6 | Recovery rate in $\mathtt{I}_2$ |
| $\beta_3$ | 1/8 | Recovery/death rate in $\mathtt{I}_3$ |
| $\gamma$ | $\approx 0.394$ | Infection/connectivity rate for ODE model ($R_0 = 2.5$). |

We refer to [76] for clinical justification of $\mu_j$, $\beta_j$.

continuous-time Markov chain (CTMC) [26] where the labeling changes randomly over time. We use the compartments described in Fig 2. Nodes change their compartment following exponentially distributed residence times corresponding to specific instantaneous rates. For the transition from *susceptible* to *exposed*, the rate depends on the neighborhood of the node (cf. Fig 2a). We consider two cases: (i) all nodes have the same infection rate $\lambda$ and (ii): each node $i$ has an individual infection rate $\lambda_i$, sampled from a probability distribution with density $v$. We start with the former case.

**Case (i): Homogeneous infectiousness.** Each $S - I_1$ can transmit an infection with rate $\lambda$. If the infected node is in compartment $I_2$ or $I_3$ the infectiousness decreases (e.g., because people stay home) and is given by $\lambda/z$. Note that we use exponentially distributed residence times which are potentially less realistic than, for instance, beta-distributions [76], but these relate directly to ODE models. Hence, observed differences in the dynamics can be attributed to the connectivity/stochasticity not the shape of the residence time.

$R_0$ is defined as the expected number of neighbors that a randomly chosen patient zero infects in a susceptible population, thus $R_0$ cannot be larger than the mean degree (number of neighbors). In the case of a fixed infection rate $\lambda$, fixing $k_{mean}$ also determines $R_0$. We can approximate $R_0$ as shown in Fig 3. We use that each infection happens independently and approximate $R_0 \approx p_I\, k_{mean}$ where $p_I$ denotes the probability that patient zero infects a random neighbor (while potentially transitioning to $I_2$, $I_3$). The approximation comes from the fact that an already infected neighbor can infect another neighbor of patient zero violating the independence assumption, rendering this an over-approximation. Note that $p_I$ is conceptually similar to the *secondary attack rate* in a completely susceptible population. We construct the networks such that $k_{mean} = 8$ (except for the complete graph where $k_{mean}$ is the number of nodes minus one). Like in the ODE-approach, we fix $R_0 = 2.5$ and determine $\lambda = 0.0706$ (cf. Fig 3).

**Case (ii): Individual differences in infectiousness.** In the case of individually varying infectiousness, we associate each node $i$ with infection rate $\lambda_i$ that is drawn from a distribution with density $v$. Again, our goal is to introduce variation while keeping the population mean. Hence, we construct $v$ such that $\lambda = E[\lambda_i] = 0.0706$. We define $R_0^i$ as the node-dependent basic reproduction number when the infection starts in node $i$. Moreover, we define the node-independent basic reproduction number as the corresponding unweighted mean $R_0 = E[R_0^i]$. Interestingly, different $v$ (with the same mean) can lead to different $R_0$. Theoretically, this follows from the computation of $p_I$ which is now based on an integral over $v$. In the next section, we

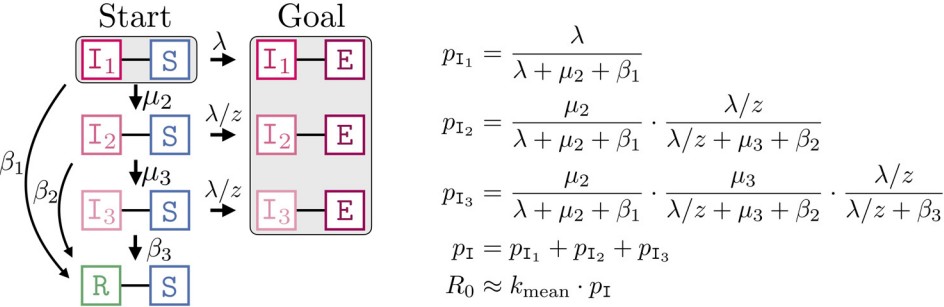

**Fig 3. Computation of $R_0$ for fixed $\lambda$. Right**: The probability that an $I_1 - S$ edge transmits the infection, $p_I$, is the sum of $p_{I_1}$: probably that the infection is transmitted while the infected node is in $I_1$; $p_{I_2}$: probability that $I_1$ transitions to $I_2$ (before transmitting the infection) and transmits while in $I_2$; and $p_{I_3}$: probability that the infection happens in $I_3$ (and not earlier). For individually varying $\lambda_i$, $R_0 \approx k_{mean} E[p_I]$ is based on an integral over $v$. **Left**: Representation of $p_I$ as a reachability probability (from *Start* to *Goal*) in a CTMC.

set $v$ to be an exponential distribution and study the resulting changes in the dynamics. A key takeaway of our study is that increasing the variance in the degree distribution does not change $R_0$, increasing the variance in the individual infectiousness does so (in fact, it decreases $R_0$). For the evaluation, we use an exponentially distributed $\lambda_i$ with mean 0.0706. That is, $v(x) = \hat{\lambda}e^{-\hat{\lambda}x}$ with $\hat{\lambda} = 1/0.0706$.

## Human-to-human contact networks

We test different types of contact networks that highlight different characteristics of real-world human-to-human connectivity. To this end, we describe the contact networks using random graph models. In each simulation, we create a specific realization (variate) of such a random graph model. A schematic visualization of example networks is provided in Fig 4. We use a **complete graph** (each possible pair of nodes is connected) as a baseline to study the evolution of the epidemic when no contact structure is present. Thereby, we can mimic the effects of stochasticity and variation in infectiousness while keeping the simulation as close as possible to the assumptions underlying the ODE. We use **Power-law Configuration Model** networks to study the effects of hubs (potential super-spreaders). These networks are—except from being constrained on having power-law degree distribution—completely random. The power-law degree distribution is omnipresent in real-world networks and entails a small number of nodes with a very high degree. We fix the minimal degree to be two and choose the power-law parameter numerically such that the network admits the desired mean degree. We also test a synthetically generated **Household** network that was loosely inspired by [88]. Each household is a clique, the edges between households represent connections stemming from work, education, shopping, leisure, etc. We use a geometric network to generate the global inter-household structure. The household size is 4. In the case of $k_{mean} = 8$, each node has 3 edges within its household and (on average) 5 outgoing edges. We also compute results for **Watts–Strogatz** (WS) random networks. They are based on a ring topology with random re-wiring. Each node has exactly $k_{mean}$ neighbors. We use a small re-wiring probability of 5% to highlight the locality of real-world epidemics.

Apart from the baseline (complete graph), we use specifically these three network models because they are well-studied in literature and very different in their respective global properties. Moreover, they all encode important properties of human-to-human connectivity like hubs (power-law), small-worldness (power-law and WS) and tightly connected household structures.

## Dispersion in networks

Given a set of independent simulation runs, we measure dispersion by analyzing the empirical offspring distribution at day $t$ (averaged over all nodes). Specifically, we consider the offspring

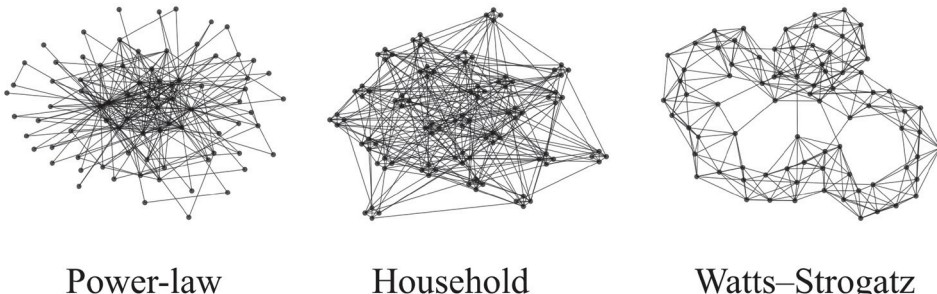

Power-law          Household          Watts–Strogatz

**Fig 4. Schematic visualizations of random graph models with 80 nodes and $k_{mean} = 8$.**

distributions of the nodes that were exposed within day $t$ (the actual secondary infections may happen later). We also perform a discretization of time over intervals of one day. We quantify dispersion in three ways:

- **CoV**: Together with the mean of the offspring distribution $R_t$, we report the *coefficient of variation* (CoV), that is the ratio of the standard deviation to mean. The CoV is a widely-used measure of dispersion of a probability distribution.

- **Top-k**: We explicitly report how many new infections within day $t$ are caused by which fraction of infected nodes (e.g., 80% of the new infections are caused by 20% of the nodes). We report which fraction of new infections can be traced back to (the most harmful) 10%, 20%, and 30% of infected nodes.

- **Offspring**: We report the fraction of nodes (that were infected within day $t$) that lead to 0, 1, 2, . . . children.

Note that overdispersion inevitably indicates not only the existence of super-spreaders but also the existence of nodes that are unlikely to pass the infection at all. Like super-spreaders, these individuals might be the result of host properties (i.e., a low viral load) or connectivity differences.

## Experiments

We compare the evolution and dispersion of the four network models. We have two main experiments. In the first experiment (overview in Fig 7), we study a fixed infection rate (mimicking the case that there is only variation in the connectivity). In the second experiment (Fig 8), we additionally impose individual variation in the infectiousness $\lambda_i$. Recall that the networks (aside from the complete graph) have approximately the same density (number of edges) and that nodes approximately admit the same mean infectiousness, thereby, ensuring that the resulting differences are solely a consequence of the corresponding variation.

Figs 5 and 6 summarize some of our findings. Fig 5 visualizes the effects of adding stochasticity and individual variation to a population. Fig 6 highlights the different dynamics emerging on different contact networks.

Python code is available at www.github.com/gerritgr/Covid19Dispersion.

## Setup

For each network, we perform 1000 simulation runs on a network with 1000 nodes. Networks are generated such that the mean degree is approximately eight. For network models where we cannot directly control $k_{mean}$, we start by generating sparse networks and increase the density until $k_{mean}$ has the desired value. In each simulation run, we start with three randomly chosen infected nodes in $I_1$ (to reduce the likelihood of initial instantaneous die-outs). The ODE (cf. Fig 6) starts with a value of 3/1000 in $I_1$. The number of simulation runs is enough to estimate the mean fractions (and the standard deviations) corresponding to each compartment with high accuracy (confidence intervals are not shown but would be barely visible anyway). The number of 1000 nodes was used for practical reasons, however, increasing the network size preserves the qualitative characteristics of the dynamics.

## Quantities of interest

We characterize epidemics in terms of the evolution of population fractions, that is, mean fraction of nodes in compartment $S$, $I_1$, $R$ for each time point $t$ (the remaining compartments evolve approximately proportional to $I_1$, thus, we leave them out for clarity). This evolution

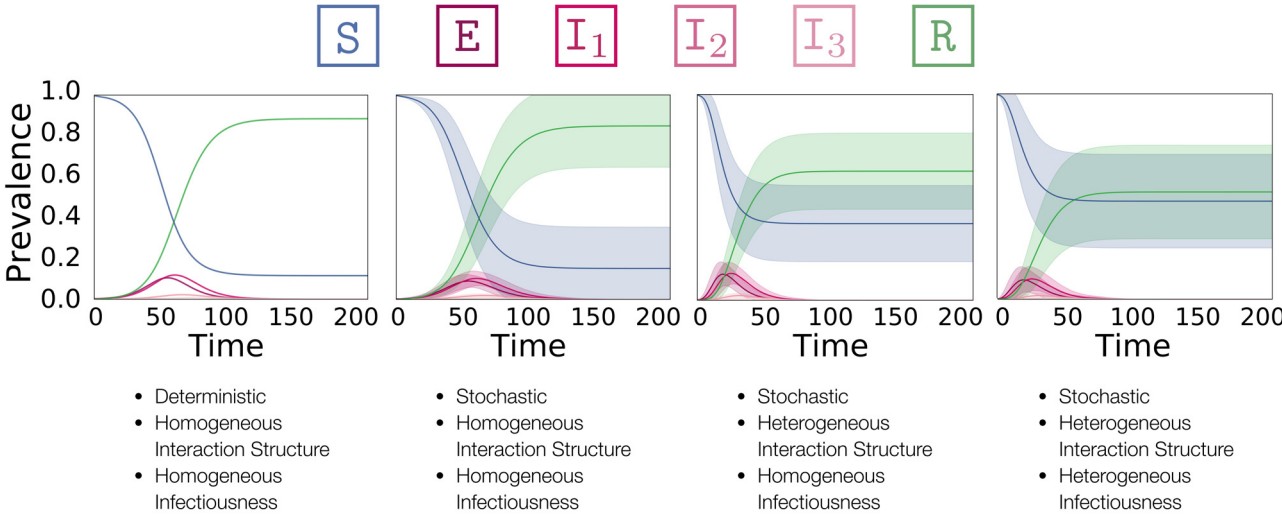

**Fig 5. Overview of how population heterogeneity shapes an epidemic. f.l.t.r.**: ODE model, a complete graph, a power-law network, and a power-law network with exponentially distributed infectiousness. The mean fraction of the population in each compartment at each point in time is shown. Shaded areas indicate standard deviations, not confidence intervals.

informs about the time point and the height of the infection peak and the final epidemic size (or HIT) that is equivalent to the (mean) fraction of recovered nodes when the epidemic is over (which is mostly the case at 200 time units). Note that the final death toll is proportional to the final epidemic size.

Moreover, we study the effective reproduction number $R_t$ (2nd row in Figs 7 and 8). We define $R_t$ to be the average number of secondary infections for a node that got exposed at day $t \in \mathbb{N}_{\geq 0}$ (we discretize time for this purpose). Thereby, we also report an empirical $R_0$ based on the three initially infected nodes that diverges slightly from the theoretical $R_0$ in Table 1. Dispersion is quantified using the three techniques explained in Section 1 (2nd to 4th row in Figs 7 and 8).

### Experiment 1: Network-based connectivity heterogeneity

Results for a fixed λ on different network types are shown in Fig 7. In most simulation runs, the power-law dynamics admits a very early peak and the epidemic dies out early with a

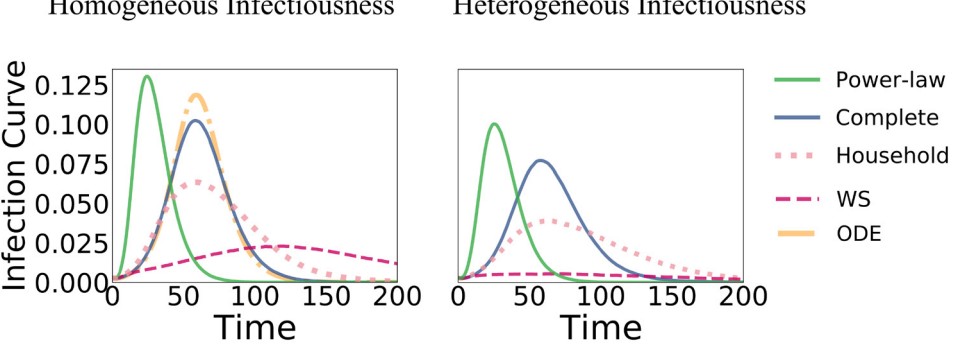

**Fig 6. Fraction of nodes in I₁ (y-axis) over time Left**: Fixed infection rate λ. **Right**: Node-based infection rate $\lambda_i$ drawn from an exponential distribution. Note the large difference between the two evolutions on the Watts–Strogatz (WS) networks.

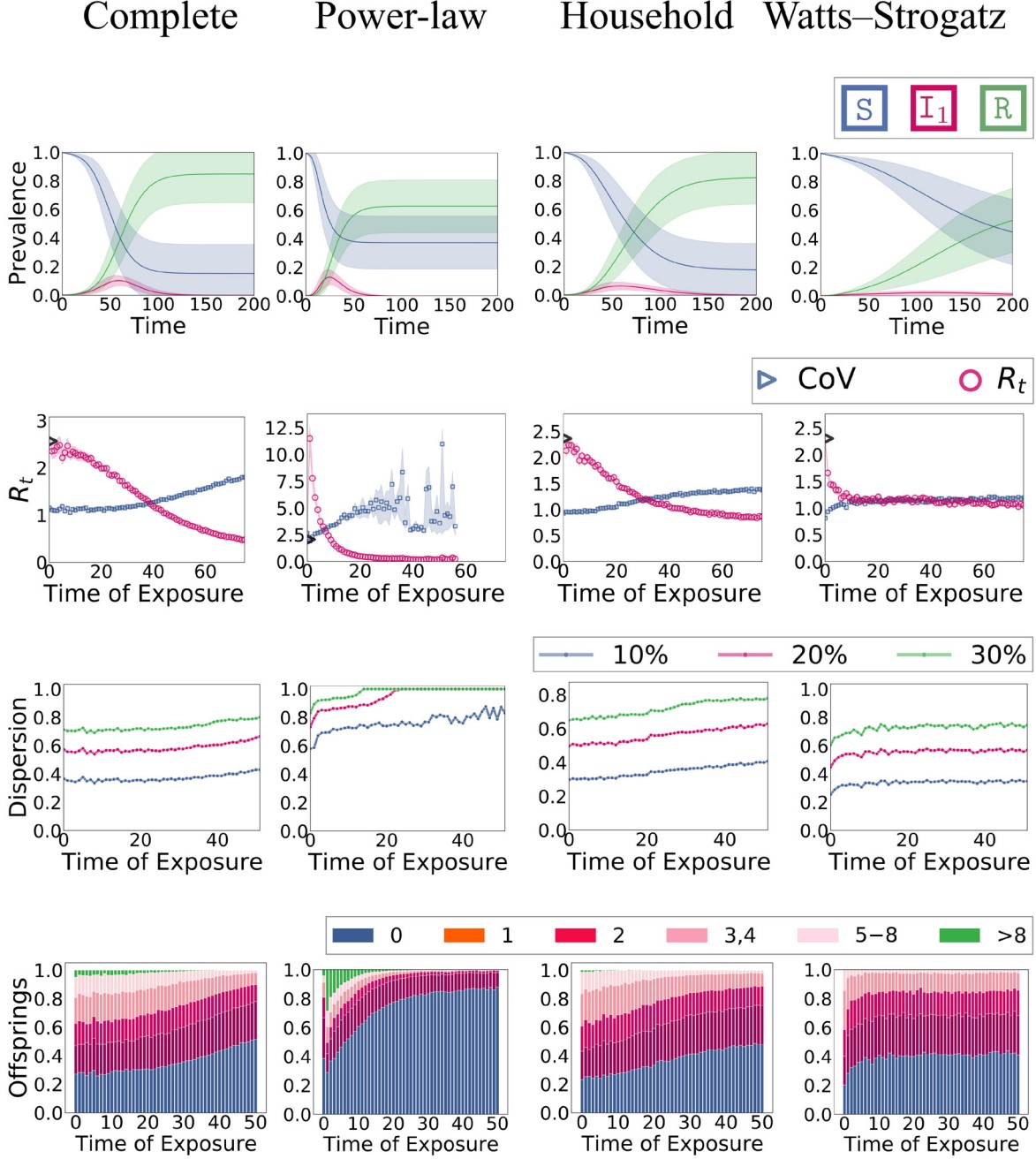

**Fig 7. Experiment 1: Epidemic dynamics of different network types. Row 1**: Evolution in terms of mean fractions (and standard deviation) in each compartment over time. **Row 2**: Effective reproduction number $R_t$ (the empirical $R_0$ is shown as a black triangle) and coefficient of variation of the offspring distribution (with 95% CI, note a significant amount of noise in the power-law case). **Row 3**: Top-$k$ plots: The fraction of new infections that can be attributed to a particular fraction of infected nodes. **Row 4**: Characterization of the offspring distribution in terms of the fraction of nodes that cause a specific number of secondary infections.

comparably small final epidemic size. This effect can directly be attributed to the hubs that get infected very early (because of their high centrality) and jump-start the epidemic. In contrast, in household networks—and even more so in WS networks—the infection peak is lower and happens at a later time point. This is no surprise as the connectivity in both networks imposes

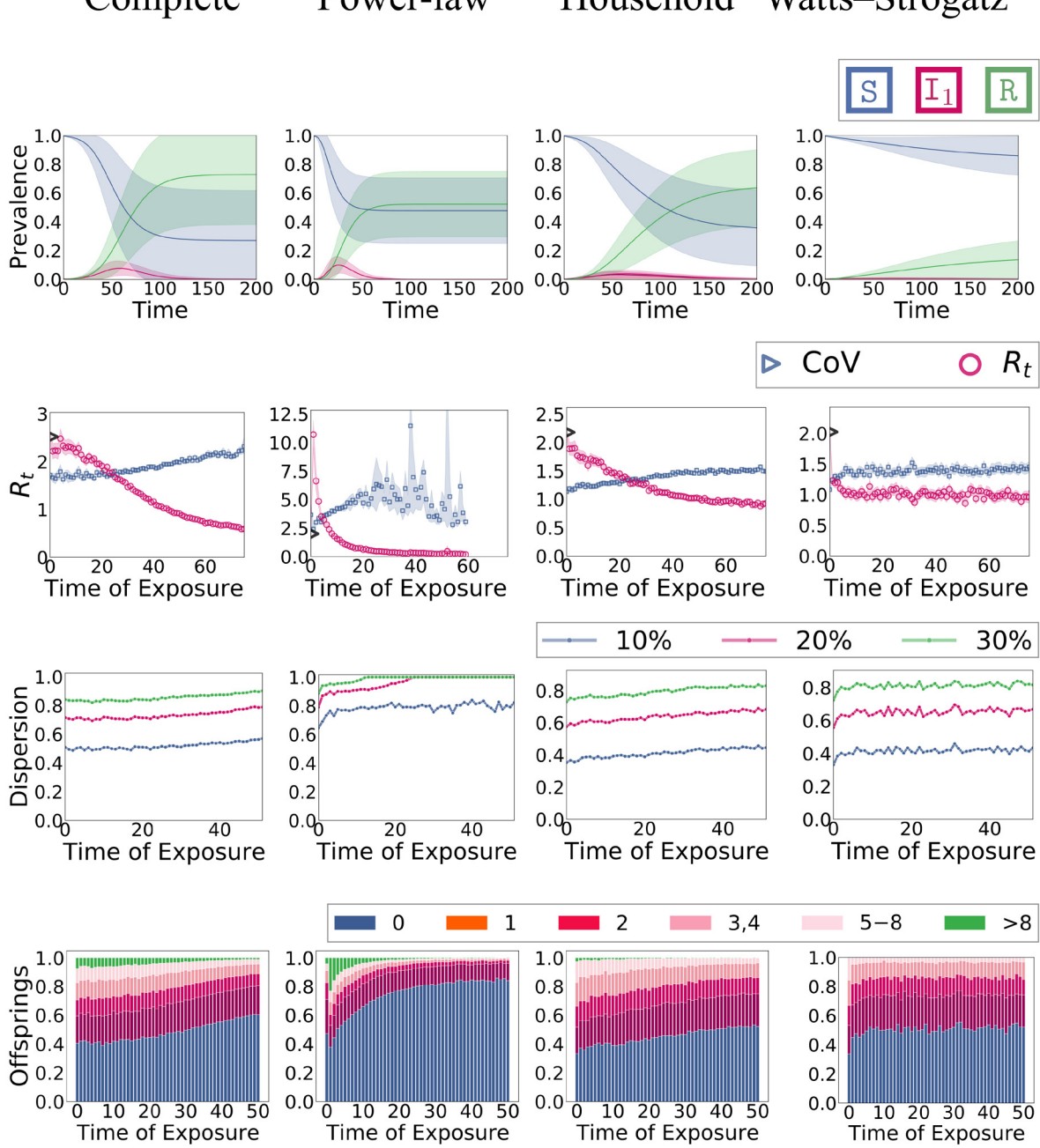

**Fig 8. Experiment 2: Epidemic dynamics with individual infectiousness $\lambda_i$. Row 1**: Evolution in terms of mean fractions in each compartment over time **Row 2**: Effective reproduction number $R_t$ and coefficient of variance of the offspring distribution. **Row 3**: Top-$k$ plots: what fraction of new infection can be attributed to which fraction of infected nodes. **Row 4**: Characterization of the offspring distribution in terms of what fraction of nodes have how many offspring (infect how many neighbors).

a level of locality that slows down the propagation. For better visibility, the differences in the infection curve (based on $I_1$) are summarized in Fig 6. We also see that the complete graph behaves very similarly to the ODE model.

The effective reproduction $R_t$ starts from around 2.5 (the theoretical overapproximation) and decreases in most cases monotonically. The exception is again the power-law network

where hubs cause a huge jump of $R_t$ in the first day from 2.5 to around 12. This jump is also reflected in the dispersion measure, most noticeable in the offspring plot (4th row). The power-law topology generally admits a higher dispersion than the other networks. For instance, the fraction of nodes with zero offspring is much higher. Moreover, on most days, the top 20% of the infected nodes account for more than 80% of the new infections which would roughly fit the estimations for COVID-19 in a typical population. In the other network types, there is less temporal variation of the dispersion. The dispersion is the lowest in the WS networks which is unsurprising as all nodes have degree 8 which provides an upper bound to the offspring number. Generally speaking, we see that dispersion can be measured robustly using the empirical offspring distribution.

Note that measuring the dispersion becomes increasingly difficult over time for the power-law network. The reason for this is that the epidemic tends to die out early with high probability. Thus, the dispersion is estimated on an comparatively small amount of samples. At the same time, the variance of the distribution is comparably high. This leads to a noticeable amount of noise.

### Experiment 2: Individual differences in infectiousness

In this experiment, we draw in each simulation run for each node $i$ a random $\lambda_i$ that is distributed according to $v$. Here we use an exponential distribution. The effects on the evolution and dispersion are reported in Fig 8. In all networks, the epidemic becomes "weaker" in terms of final epidemic size and infection peak height (see also Fig 6). This effect is strongest in the WS network (where the epidemic dies out almost immediately) and weakest in the complete graph. This is also mirrored in the difference of $R_0$ compared to Experiment 1 (as explained in Fig 3, the relationship between $\lambda$ and $R_0$ is now non-linear). The complete graph leaves $R_0$ almost unchanged (i.e., around 2.5) while it goes down to around 2.0 in the WS network. Effects on the household and power-law network are less drastic but still evident.

Regarding the dispersion, the differences to Experiment 1 are expected. The variance in the empirical offspring distribution generally increases. Interestingly, this happens in all networks by roughly the same amount regardless of whether the dispersion was high or low in the first case. We can also consistently see the change in dispersion in all three dispersion measures, but it is especially evident in the top-$k$ plots (3rd row). It is also interesting to see that all networks admit a characteristic signature in the histogram of the offspring distribution (4th row). The infection rate variation shifts these plots (in particular, because the number of nodes without offspring increases) but they still entail a clear distinction between networks.

We also tested uniform and gamma distributions for $v$ (results not shown), and found that the epidemic generally becomes weaker with higher variance. We expect that this is due to an increased likelihood of local and global die-outs.

### Discussion

The two experiments show that heterogeneous interaction structures and variations in infectiousness strongly influence key quantities of an epidemic. However, there are important differences between the sources of variations:

- The existence of hubs in the network can cause $R_t$ to increase, variations in $\lambda$ generally do not cause this behavior.

- Different networks with the same $k_{\mathrm{mean}}$ and a fixed $\lambda$ will (approximately) admit the same $R_0$. However, choosing densities $v$ of different form (while keeping the mean of the distribution fixed) changes $R_0$.

- Allowing infectiousness to vary between individuals generally weakens the epidemic's impact. Allowing infectiousness to vary between individuals can make some aspects of the epidemic worse (e.g., height of the infection peak in power-law networks).

- $\lambda$ has the weakest influence when the interaction structure is homogeneous (i.e., on a complete graph) and the strongest when the interaction structure is based on locality (the average distances in the graph increase) as in the Watts–Strogatz network.

- Varying $\lambda$ increases the stochasticity (e.g., the variance of the number of infected nodes at any given point in time) of the epidemic.

- The interaction structure has a large influence on the dispersion. Individual infectiousness variations induces a smaller but consistent increase of dispersion.

- Hubs influence how the dispersion changes over time. Infectiousness variations increase the dispersion consistently for all time points.

Our results underline that networks are a feasible tool to encode a wide variety of different features of a population's interaction structure. Generally speaking, it is not surprising that some networks support the formation of epidemics better than others. To some extent, this has been studied in terms of the epidemic threshold of graphs [89]. However, the variety of the influence of the networks and the interplay between heterogeneity in the degree of infectiousness and dispersion is remarkable and, to our knowledge, underexplored in literature.

There are even further possibilities to adjust population heterogeneity, e.g., by adding non-Markovian residence times in the compartments, by varying the remaining transition rates, or by imposing more temporal variability in infectiousness. Our results already show that models, based on point estimators of population averages (i.e., most mean-field ODE models), are not adequate for analyzing or predicting the dynamics of an epidemic.

Regarding the dispersion, we see that none of the considered network structures by itself leads to a dispersion where 80% of the infections are caused by only 15% of the infected nodes (at least not right from the beginning, in the power-law graph this point is approximately reached within a few days). However, the differences between networks are remarkable. From branching process theory, it is known that a higher dispersion increases the die-out probability [12]. Generally, this effect also holds for networks. For a fixed network, increasing dispersion by using a probabilistic infection rate does, in fact, increase the die-out probability. However, the network topology strongly modulates the strength of this effect.

Conclusively, we find that in most cases population diversity makes an epidemic less harmful but increases the dispersion and the variability of the evolution. Hubs in the contact networks are an exception to this rule. These are drivers of the epidemic as they become infected very early and infect many others. This distinguishes them from very infectious people (due to a high viral load) with an average number of contacts who also potentially infect many others. However, a high infectiousness alone does not make them more likely to be infected early enough (i.e., on average earlier than other nodes) to cause an early explosive surge of the epidemic. Hubs also highlight that the effective reproduction number can change significantly while the environmental conditions remain the same simply because the prevalence shifts towards highly connected individuals in the beginning.

Considering that an exponentially distributed $\lambda_i$ can be considered a fairly strong assumption about individual differences, our work can—with necessary caution—be seen as further evidence that the network structure plays a more important role for the dispersion than individual viral load variability.

Transferring these characteristics to NPIs, our work indicates that reducing long-range connections (e.g., by corresponding mobility restrictions) and keeping degree-variability small (to avoid hubs) are particularly effective control strategies. Reducing mobility seems to be especially effective for overdispersed epidemics. Note that the differences between the WS networks (that admit a high level of locality) and the other networks become even more evident when we vary infectiousness. This can be explained by the observation that in overdispersed epidemics the virus has to be introduced to a susceptible population multiple times before an outbreak becomes likely.

## Conclusions

We tested the influence of heterogeneity in the population's interaction structure and degree of individual infectiousness on the dynamical evolution of an epidemic. We find that the dynamics depends strongly on these properties and that there is an intriguing interplay between these two sources of variation. Our work also highlights the role of small-worldness, local die-outs, and super-spreaders for an epidemic.

Naturally, mathematical modeling is based on assumptions and abstractions. However, heterogeneity seems to be particularly vital and excluding it should only be done with great caution. It is particularly difficult to capture population heterogeneity in the widely-used class of ODE models. This is due to their inherent homogeneity assumptions w.r.t. each compartment. Although effects such as overdispersion can be modelled to some extent using this paradigm [90], the complex interplay of varying infectiousness and connectivity remains mostly elusive for such models. Discussing epidemics in terms of population averages may not adequately reflect the complexity of the emerging dynamics.

On a high-level, this work highlights limitations of certain model classes and shows that subtle differences in assumptions can make important differences. For future work, it would be interesting to study the effects of heterogeneity and their implications for NPIs empirically. Moreover, we plan to test implications of further sources of heterogeneity, for instance regarding compliance with NPIs, susceptibility to infections, or whether people tend to meet indoors or outdoors.

In a broader spectrum, population heterogeneity is only one aspect that may cause models to perform much worse in the real-world than one might expect. This simulation study is a reminder that models are prone to hidden assumptions, and that we should be cautious with their interpretation.

## Author Contributions

**Conceptualization:** Gerrit Großmann, Michael Backenköhler, Verena Wolf.

**Formal analysis:** Gerrit Großmann.

**Funding acquisition:** Verena Wolf.

**Investigation:** Gerrit Großmann.

**Methodology:** Gerrit Großmann, Michael Backenköhler, Verena Wolf.

**Project administration:** Verena Wolf.

**Resources:** Verena Wolf.

**Software:** Gerrit Großmann.

**Writing – original draft:** Gerrit Großmann, Michael Backenköhler, Verena Wolf.

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
