## [Decision Letter · Decision Letter 0]

30 Apr 2021

PONE-D-21-09416

Why ODE models for COVID-19 fail: Heterogeneity shapes epidemic dynamics

PLOS ONE

Dear Dr. Grossmann,

Thank you for submitting your manuscript to PLOS ONE. After careful consideration, we feel that it has merit but does not fully meet PLOS ONE’s publication criteria as it currently stands. Therefore, we invite you to submit a revised version of the manuscript that addresses the points raised during the review process.

Reviewers have commented on your manuscript and recommend a revision. For your guidance, we append reviewers' comments. You have an opportunity to improve the paper; please submit a carefully revised manuscript and a list of your responses.  

We look forward to receiving your revised manuscript.

Kind regards,

Vygintas Gontis, Ph.D.

Academic Editor

PLOS ONE

Journal Requirements:

"This work was partially supported by the DFG project MULTIMODE (DFG 391984329)

Otherwise, the author(s) received no specific funding for this work."

Reviewers' comments:

Reviewer's Responses to Questions

**Comments to the Author**

1. Is the manuscript technically sound, and do the data support the conclusions?

Reviewer #1: Yes

Reviewer #2: Partly

2. Has the statistical analysis been performed appropriately and rigorously? 

Reviewer #1: Yes

Reviewer #2: Yes

3. Have the authors made all data underlying the findings in their manuscript fully available?

Reviewer #1: Yes

Reviewer #2: Yes

4. Is the manuscript presented in an intelligible fashion and written in standard English?

Reviewer #1: Yes

Reviewer #2: Yes

5. Review Comments to the Author

Reviewer #1: The authors report on stochastic simulations employing a translated COVID-19 ODE model for distinct contact networks and discuss the impacts of their findings in terms of the population's interaction structure and the degree of individual infectiousness. The authors present a short introductory review of the literature regarding the modeling of the COVID-19 spread and some results regarding the role played by population heterogeneity in the spread of such a disease. Essentially, heterogeneity and dispersion can be considered as key words in the present work. Below I raise some aspects I find relevant for clarification:

1) The title of the manuscript seems misleading to me since the authors make use of a translated ODE model to investigate different contact networks and study COVID-19 spread. It seems that the authors may have intended to mention that the ODE models alone are incomplete to describe the COVID-19 spread. I suggest that the authors make a brief comment about this. In the present version, it sounds like ODE models are useless.

2) The authors incorporate in their analysis the so-called "super-spreaders". What about the "super non-spreaders", i.e., people who have dramatically reduced contacts with others? This may be the case for most elderly people. Does the employed model take this into account? This may be related to age, since young people contribute to a higher number of infections than older people, see, for instance, Science 371, 1336 (2021). I consider that this point deserves a brief discussion in the manuscript.

3) In line 168, the authors write:

"... In contrast to [71], we merge dead and recovered stages to a single removed stage, as both do not influence the infection dynamics further (we assume immunity after recovery). ..."

The authors should be careful in stating that recovered people do not further influence the infection dynamics. Recovered people, as the authors assume, are immune to some extent (not considering, for instance, distinct variants of the virus) and can act as suppressors of the creation of new infection branches in contact networks. Thus they may significantly contribute to mitigate the COVID-19 spread. Also, the recovered (immune) and vaccinated people can give rise, to some extent, to the herd-immunity widely discussed in the literature. I suggest that the authors make a short discussion about this in the manuscript.

4) Why do the authors have specifically employed a SIR-type approach on the ODE model (line 165) instead of, for instance, a SEIS- or a SEIRS-type model? This question is connected with the point raised in 3).

5) In the caption of Fig.7, the authors write:

"... note a significant amount of noise in the power-law case). ..."

Such noise in the power-law case can also be observed in Fig.8. What is the origin of such a noise? What does this noise mean and why it is not observed for the other cases? I suggest that the authors comment on this aspect.

6) What is the justification for why do the authors have specifically employed the power-law, household, and Watts-Strogatz contact networks? I suggest that the authors make this more explicit in the manuscript.

7) In line 414, the authors write:

“… Specifically, the widely-used class of ODE models cannot accurately capture population heterogeneity. Naturally, mathematical modeling is based on assumptions and abstractions. …”

As a matter of fact, ODE models are not based on “assumptions and abstractions”. Instead, one has well-established ODE models, which can give us relevant information about the mitigation of the Covid-19. In this respect, the authors should reformulate such sentence and cite the following references:

K.E. Nelson, C.M. Williams, Infectious Disease Epidemiology: Theory and Practice, Jones & Bartlett Publishers, Burlington, 2014.

J.C. Frauenthal, Mathematical Modeling in Epidemiology, Springer Science & Business Media, New York, 2012.

S. Ma, Y. Xia, Mathematical Understanding of Infectious Disease Dynamics, World Scientific, Singapore, 2009.

F. Brauer, P.D. Driessche, J. Wu, Lecture Notes in Mathematical Epidemiology, Springer, Berlin, 2008.

Physica A: Statistical Mechanics and its Applications 573, 125963 (2021).

8) Minor typos throughout the manuscript should be correct, such as:

- "... are summarized..." instead of "... a summarized..." in line 98;

- "... in each disease stage ..." instead of "... in each disease sage ..." on the caption of Fig.2;

- "... For the transition from susceptible to exposed, ..." instead of "... For the transition from susceptible top exposed, ..." in line 185.

8) The authors should state in a more clear-cut way what is new in the present work based on similar publications employing network models for COVID-19, since after the COVID-19 outbreak, an uncountable number of articles have been published employing distinct models aiming to describe the COVID-19 spread.

Should the authors clarify the points I raised, I would recommend the manuscript for publication.

Reviewer #2: In the manuscript "Why ODE models for COVID-19 fail: Heterogeneity shapes epidemic dynamics" PONE-D-21-09416, the authors investigate how variations in the connectivity and infectivity of individuals affects the evolution of pandemics and in particular features such as the evolution of the effective reproduction number, the total number of infections, and the heard immunity threshold. To this end, the authors investigate the evolution of pandemics on networks with varying connectivity characteristics and with either constant or individually varying infection rates and compare these with results obtained by an ODE model with the same compartments. To establish a meaningful comparison, either between different networks or between the network-based approaches and the ODE model, the authors choose their parameters according to certain conditions, which in general require that the basic reproduction number ($R_0$) is equal between all variations of the models. The experiments presented reveal that variations in connectivity and the distribution of infection rates among individuals affects the evolution of the pandemic as measured by the time evolution of a number of quantities of interest defined in the manuscript, and in particular varies from the ODE model.

The methodology used by the authors is appropriate, the analysis is thorough, and the manuscript is well written and clear. The results are well presented and the figures are discussed adequately. However, before I can recommend publication, I would suggest clarifying some points I have identified, or alternatively revise the manuscript as explain in the long report of the attached PDF

6. PLOS authors have the option to publish the peer review history of their article (what does this mean?). If published, this will include your full peer review and any attached files.

Reviewer #1: No

Reviewer #2: No

---

## [Author Response · Author response to Decision Letter 0]

21 Jun 2021

We provide a "Response to Reviewers" letter that contains our answers to the reviewers.

---

## [Decision Letter · Decision Letter 1]

6 Jul 2021

Heterogeneity matters: Contact structure and individual variation shape epidemic dynamics

PONE-D-21-09416R1

Dear Dr. Grossmann,

We’re pleased to inform you that your manuscript has been judged scientifically suitable for publication and will be formally accepted for publication once it meets all outstanding technical requirements.

Kind regards,

Vygintas Gontis, Ph.D.

Academic Editor

PLOS ONE

Additional Editor Comments (optional):

Reviewers' comments:

Reviewer's Responses to Questions

**Comments to the Author**

1. If the authors have adequately addressed your comments raised in a previous round of review and you feel that this manuscript is now acceptable for publication, you may indicate that here to bypass the “Comments to the Author” section, enter your conflict of interest statement in the “Confidential to Editor” section, and submit your "Accept" recommendation.

Reviewer #1: All comments have been addressed

2. Is the manuscript technically sound, and do the data support the conclusions?

Reviewer #1: Yes

3. Has the statistical analysis been performed appropriately and rigorously? 

Reviewer #1: Yes

4. Have the authors made all data underlying the findings in their manuscript fully available?

Reviewer #1: Yes

5. Is the manuscript presented in an intelligible fashion and written in standard English?

Reviewer #1: Yes

6. Review Comments to the Author

Reviewer #1: I thank the authors for patiently go through all of my suggestions. All of my queries/concerns regarding the manuscript has been clarified diligently.

The manuscript has been noticeably improved and seems pretty complete to me. I wish the authors luck for a good publication in PLOS ONE!

7. PLOS authors have the option to publish the peer review history of their article (what does this mean?). If published, this will include your full peer review and any attached files.

Reviewer #1: No

---

## [Editor Report · Acceptance letter]

9 Jul 2021

PONE-D-21-09416R1 

Heterogeneity matters: Contact structure and individual variation shape epidemic dynamics 

Dear Dr. Grossmann:

I'm pleased to inform you that your manuscript has been deemed suitable for publication in PLOS ONE. Congratulations! Your manuscript is now with our production department. 

Kind regards, 

on behalf of

Dr. Vygintas Gontis 

Academic Editor

PLOS ONE